# Event Discovery for History Representation in Reinforcement Learning

## Abstract

Environments in Reinforcement Learning (RL) are usually only partially observable. To address this problem, a possible solution is to provide the agent with information about past observations. While common methods represent this history using a Recurrent Neural Network (RNN), in this paper we propose an alternative representation which is based on the record of the past events observed in a given episode. Inspired by human memory, these events describe only important changes in the environment and, in our approach, are automatically discovered using self-supervision. We evaluate our history representation method using two challenging RL benchmarks: some games of the Atari-57 suite and the 3D environment Obstacle Tower. Using these benchmarks we show the advantage of our solution with respect to common approaches.

## 1 Introduction

Deep Reinforcement Learning (RL) algorithms have been successfully applied to a range of challenging domains, from computer games (Mnih et al., 2013) to robotic control (OpenAI et al., 2018). These approaches use a Neural Network (NN) both to represent the current observation of the environment and to learn the agent's optimal policy, used to choose the next action. For instance, the state observation can be the current game frame or an image of the robot camera, and a Convolutional Neural Network (CNN) may be used to obtain a compact feature vector from it.

However, often RL environments are only partially observable and having a significant representation of the past may be crucial for the agent (Kapturowski et al., 2019). For instance, in the Atari 2600 Pong game, the state can be represented as two consecutive frames. In this way, the agent can determine both the position of the ball and the direction of its movement. More complex partially observable domains require a longer state history input to the agent. For instance, when navigating inside a 1-st person-view 3D labyrinth, the agent obtains little information from the current scene observation and needs several previous frames to localise itself.

A common solution to represent the observation history is based on the use of Recurrent Neural Networks (RNNs), where the RNN hidden-layer activation vector is input to the agent, possibly together with the current state observation (Mnih et al., 2016). However, RL is characterized by highly nonstationary data, which makes training unstable (Schulman et al., 2016) and this instability is exacerbated when a recurrent network needs to be simultaneously trained to extract the agent's input representation. We show in Sec. 5.2 that in some common cases an RNN-based history representation struggles to improve the agent's results over an input representation composed of only the instantaneous observations. In this paper we propose a different direction, in which the agent's observation history is represented using a set of discrete *events*, which describe important changes in the state of the world (Orr et al., 2018). Environment-specific events are automatically discovered during training by clustering past observations and are then used as landmarks in our history representation. Intuitively, this is inspired by the common human behaviour: when making decisions, humans do not keep detailed visual information of the previous steps. For instance, while navigating through the halls of a building, it is sufficient to recall a few significant landmarks seen during the walk, e.g. specific doors or furniture.

Following this idea, in this paper we propose an Event Discovery History Representation (EDHR) approach, composed of 3 iterative stages: *experience collection, policy optimisation* and *event discovery*. We discover events by clustering past state observations. In more detail, we maximize the

Mutual Information (MI) between the latent representations of temporally-close frames to cluster the frame semantics. This is a form of *self-supervision*, in which no additional annotation is needed for the event discovery: the higher the *predictability* of one frame with respect to the other, the larger the semantic information shared by the two frames (van den Oord et al., 2018; Hjelm et al., 2019; Anand et al., 2019; Ji et al., 2019).

Once clusters have been formed, the probability distribution of a given frame $F_t$ for the set of current clusters is used as the semantic representation of the state observation at time $t$ and is recorded in a longer *history*. Finally, the history is input to the agent together with an *instantaneous* observation representation, obtained, following (Mnih et al., 2013), as a stack of the last 4 frames. This information is now used for policy optimisation. Note that our proposed history representation is independent of the specific RL approach. However, in all our experiments we use the PPO algorithm (Schulman et al., 2017) for the policy and the value function optimisation. The 3 stages are iterated during training, and thus past clusters can be modified and adapted to address new observations, while the agent is progressing through the environment.

We summarize the contribution of this paper below. First, we use a modified version of the Invariant Information Clustering (IIC) algorithm (Ji et al., 2019) to discover significant events in the agent's past observations. Second, we propose to replace common RNN-based history representations with a time-dependent probability distribution of the observations for the events so far discovered. We evaluate our history representation (EDHR) using the PPO algorithm on several environments of the Atari-57 benchmark (Bellemare et al., 2012) and on the 3D environment Obstacle Tower (Juliani et al., 2019), showing that it provides a significant boost with respect to the most common input representation methods. Specifically, we show that EDHR outperforms plain PPO in cases, where history can increase observability, and it outperforms RNN-based methods in several common cases, while simultaneously requiring $2\times$ less wall-clock time for training.

The source code of the method and all the experiments is publicly available at the anonymous link: https://github.com/iclr2020anon/EDHR and will be published after acceptance.

## 2    RELATED WORK

**Partially Observable Markov Decision Processes.** We consider a setting in which an agent interacts with an environment to maximize a cumulative reward signal. The interaction process is defined as a Partially Observable Markov Decision Process (POMDP) $< S, A, T, R, \Omega, O, \gamma >$ (Monahan, 1982; Kaelbling et al., 1998), where $S$ is a finite state space, $A$ is a finite action space, $R(s, a)$ is the reward function, $T(\cdot|s, a)$ is the transition function, $\Omega$ is a set of possible observations, $O$ is a function which maps states to probability distributions over observations and $\gamma \in [0, 1)$ is a *discount factor* used to compute the cumulative discounted reward. Within this framework, at each step $t$, the environment is in some unobserved state $s \in S$. When the agent takes an action $a$, the environment transits to the new state $s'$ with probability $T(s'|s, a)$, providing the agent with a new observation $o' \in \Omega$ with probability $O(o'|s')$ and reward $r \sim R(s, a)$.

POMDPs are an active research area and several approaches have been recently proposed to address partial observability. DQN (Mnih et al., 2013) is one of the earliest deep RL models directly trained using the high-dimensional observation $o_t$ as input (i.e., the raw pixels of the current frame). To deal with partial observability, the input of the model is composed of a stack of the 4 last grayscale frames, processed as 4 channels using a 2D CNN. We also use this observation representation which we call *instantaneous* to highlight the difference from the past state history.

Hausknecht & Stone (2015) replace the fully-connected layer in DQN with an LSTM (Hochreiter & Schmidhuber, 1997) layer. At each step, the model receives only one frame of the environment together with the LSTM hidden-layer value which represents the past. The idea was extended in (Igl et al., 2018) by inferring latent belief states. To simulate reduced observability and demonstrate the robustness of the method, Hausknecht & Stone (2015) introduce the flickering Atari games, obscuring the observation with a certain probability at each step. These methods focus on short-term partial observability, such as noisy observations, and approximate the state with an implicitly inferred continuous history. In contrast, our method explicitly represents each step of the past using a dictionary of discovered events, and concatenates a sequence of steps in an event-based history.

Our final goal is to augment the current state representation with important past changes of the environment.

Recurrent layers (e.g. LSTM or GRU (Cho et al., 2014)), are a common choice for modern RL architectures. The A3C algorithm (Mnih et al., 2016) showed a high mean performance with an LSTM layer. R2D2 (Kapturowski et al., 2019) is specifically focused on processing long-history sequences (80 steps with additional 40 burn-in steps) using an LSTM. However, despite its widespread, RNNs have disadvantages within an RL framework because they increase the training instability (Sec. 1). In this paper we propose an alternative history representation and we empirically show in Sec. 5.2 that our proposal outperforms RNN-based solutions in several common cases.

**Unsupervised and Self-supervised Learning.** Unsupervised and self-supervised learning aim to extract useful representations from data (e.g. images, videos) without annotation and reward signal. While common approaches are covered in recent surveys (Jing & Tian, 2019; Kolesnikov et al., 2019), below we analyse the most relevant methods to this work.

A first class of unsupervised feature-extraction methods is based on a generative encoding-decoding scheme. The methods typically minimize the reconstruction error in the pixel space, thus the representation is encouraged to capture pixel-level details, which may differ from semantics. Ha & Schmidhuber (2018) combine Variational Autoencoders (Kingma & Welling, 2014) (VAEs) with RNNs in their *World Models*. A VAE is trained to reconstruct the current frame and simultaneously produce a latent, compressed frame representation which is input to an RNN predicting the next latent feature vector. The input of the RL agent consists of the latent vector of the current observation and the hidden state of the RNN.

An alternative to a reconstruction loss in the input space is clustering in the feature space. Deep-Cluster (Caron et al., 2018) iteratively alternates between k-means clustering of the CNN features and updating the network's parameters with the pseudo-labels obtained by means of clustering.

Another important self-supervised paradigm is based on maximizing the Mutual Information (MI) between latent representations. Typically these methods rely on collecting pairs of *positive* and *negative* samples. Two positive samples should share the same semantics, while negatives should be perceptually different. A discriminator is trained to distinguish between positive and negative pairs, so computing a lower-bound on the MI of the learned latent representations. In CPC (van den Oord et al., 2018) and ST-DIM (Anand et al., 2019) positives are temporally close frames, while in IIC (Ji et al., 2019) simple data-augmentation techniques are applied on still images. In DIM (Hjelm et al., 2019) this idea is extended including *locality*, and MI is computed also between global and local representations of the same image. DIM was applied to several games of the Atari 2600 suite (Bellemare et al., 2012) in ST-DIM, however, the reported evaluation is based only the disentanglement degree of the obtained representations and does not consider the possible benefit for an RL task.

Our event discovery stage (Sec. 3) is based on IIC (Ji et al., 2019), but we use as positives temporally-close frames extracted from an agent trajectory instead of data-augmentation. Note that the main difference between our proposal and IIC is that our goal is not clustering, but clusters are used as a set of discrete events to represent the agent's history, in contrast to RNN and other continuous history representations.

## 3 EVENT DISCOVERY AND HISTORY REPRESENTATION

**Event discovery stage.** As mentioned in Sec. 1-2, we use IIC (Ji et al., 2019) for our unsupervised event discovery. While we refer the reader to the Appendix C and the original paper for the details, we show below how we use this clustering method to obtain a set of discrete events and simultaneously learn to encode our observations.

Let $o_t$ be the observation of the environment at time step $t$, presented as an image. We want to learn a representation mapping $\Phi : \Omega \to [0,1]^C$, where $Y = \{1, ..., C\}$ is a set of $C$ events and $\Phi(o_t)$ is the probability distribution of a discrete random variable $z$ which (soft-)assigns $o_t$ to the elements in $Y$ and can be defined as: $\Phi_c(o_t) = P(z = c|o_t)$ (Ji et al., 2019). $\Phi(\cdot)$ is our *unsupervised encoder* and it is based on a CNN followed by a fully-connected layer and a softmax layer (Fig. 1), parametrized by $\phi$. The clustering process in IIC is based on pairs of images randomly transformed which contain

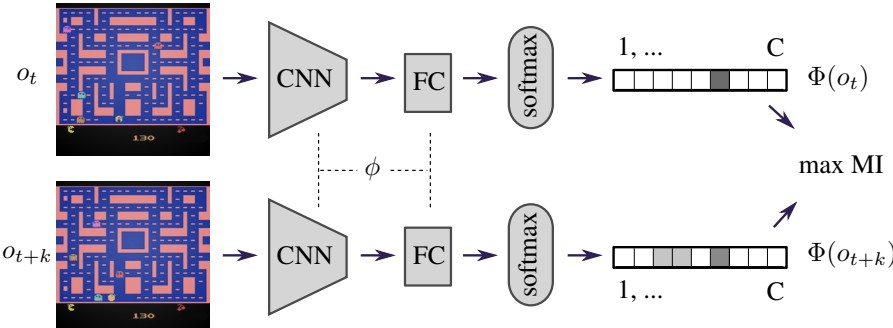

Figure 1: Event discovery. This figure shows two replicas of our *unsupervised encoder* architecture ($\Phi$) which separately map two different observations in corresponding event probability distributions. The latter are used to maximize the MI over the latent assignment variable $z$, in this way obtaining a set of events $Y$ and the supervision signal used to update the parameters ($\phi$) of $\Phi$.

a similar semantics. In our case, we use two observations extracted from the same trajectory: $o_t$ and $o_{t+k}$, where $k \sim U[1, L]$. $L$ defines a temporal translation window in which observations are assumed to mostly like have the same semantic content. Small values of $L$ (e.g. $L = 1$) reduces the uncertainty and thus improves the stability during training. On the other hand, larger values of $L$ produce a higher intra-cluster variability, in which the clustering process is forced to extract a more abstract semantics to group similar frames. This parameter can vary for different RL environments, in our experiments in Sec. 5.2 we use $L = 3$, additionally we demonstrate the effect of adjusting this parameter in Appendix B.

Following (Ji et al., 2019), clustering is performed by computing a $C \times C$ matrix whose elements represent the joint probability $P(z = c, z' = c'|o_t, o_{t+k})$ and then maximizing the MI directly using this matrix.

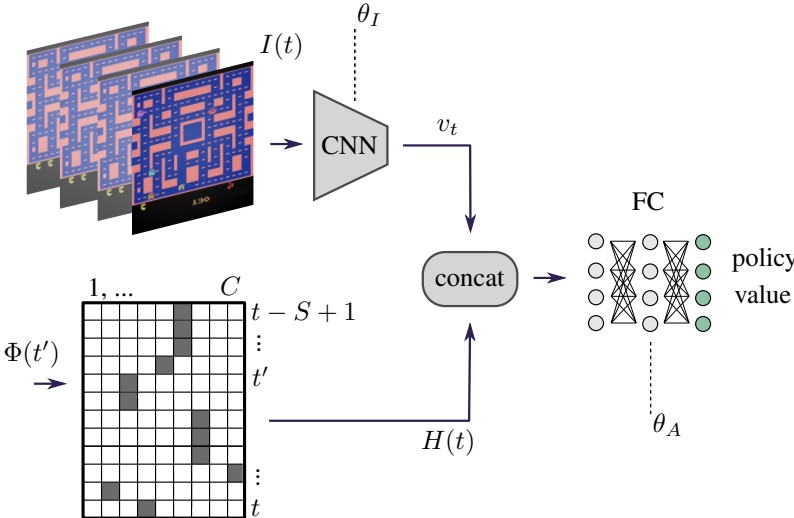

Figure 2: The full agent model and our history representation.

**History Representation and Policy Optimisation.** Once $\phi$ is updated (see Sec. 4), we use $\Phi(o_t; \phi)$ to represent an observation at time $t$ and update the agent's policy. In more detail, we represent the agent with an Actor-Critic architecture approximated with a NN $\pi$ parametrised by $\theta$. We use the PPO algorithm to optimise the policy and the value function. At time step $t$, the agent receives two types of data as input, an *instantaneous* representation $I(t)$ consisting of 4 concatenated raw-pixel, grayscale frames $I(t) = \{o_t, o_{t-1}, o_{t-2}, o_{t-3}\}$ and a *history* $H(t)$ representing past events. Specifically, for each $t' \in \{t, ..., t - S + 1\}$ let $\Phi(o_{t'}; \phi)$ be the probability distribution of $o_{t'}$ for the current set of events $Y$. Our proposed history representation is simply given

---

**Algorithm 1** Training

1 Initialize $\phi$ and $\theta$.
2 For $iteration = 1, ...N$ do:
3     For $actor = 1, 2, ...$ do:
4         For $t = 1, ..., r$ do:
            **Observation collection stage:**
5             Run $\pi_\theta(I(t), H(t))$ in the environment and get $o_{t+1}$
6             Use $\Phi(o_{t+1}; \phi)$ to compute $H(t + 1)$.
7             Collect $I(t + 1)$, $H(t + 1)$ and other PPO-specific information in a buffer $B$.
        **Policy optimisation stage:**
8         Use $B$ to update $\theta_I$ and $\theta_A$ in $n_P$ epochs
        **Event discovery stage:**
9         Use $B$ to update $\phi$ in $n_E$ epochs

---

by the concatenation of these cluster soft-assignment vectors, computed over the past $S$ steps: $H(t) = \{\Phi(o_t; \phi), ..., \Phi(o_{t-S+1}; \phi)\}$, arranged as a $S \times C$ matrix.

We empirically observed that the history matrix $H(t)$ is very sparse (Sec. 5.3), being most of the elements in $\Phi(o_t)$ close to 0 ($\Phi(o_t)$ is usually very peaked). Thus, an even more compact history representation may be produced. Anyway, with a small value of $S$ ($S = 32$ in all our experiments) and a small number of clusters $C$ (we use $C = 16$), $H(t)$ is a low-dimensional, sparse matrix which represents significant information about the recent agent's past trajectory. Importantly, in contrast with an RNN-based history representation, which may be prone to forgetting past information, in our proposed history representation $H(t)$, all the past $S$ events are stored, together with their time-dependent evolution.

$I(t)$ is input to a second CNN ($\Psi$) which extracts a vector $v_t$ representing the instantaneous visual information ($v_t$ is obtained by flattening the last CNN feature map). We call the parameters of $\Psi$ $\theta_I$, to differentiate them from the agent's proper parameters $\theta_A$ which control the policy and the value function ($\theta = [\theta_I; \theta_A]$). Finally, $v_t$ is concatenated with the (flattened) matrix $H(t)$ and input to the agent. Fig. 2 shows the full architecture of our agent. We highlight that the CNN of the unsupervised encoder ($\Phi$) is different from $\Psi$, and the two CNNs do not share parameters ($\phi \neq \theta_I$). Moreover, while the encoder is trained in an unsupervised fashion, $\Psi$, used to extract visual information ($v_t$) from $I(t)$, is trained *using the RL reward*. This emphasizes the complementary between the knowledge of the two networks, being the former based on clustering visual similarities in data which do not depend on the specific RL task, while the latter being trained as in a standard RL framework.

## 4 TRAINING

We run the agent in several parallel environments for a predefined number of steps (rollout size $r$). The actions are chosen according to the current policy and encoder (*observation collection stage*). Next, using the collected experience, we update $\theta$ for $n_P$ epochs (*policy optimisation stage*) and $\phi$ for $n_E$ epochs (*event discovery stage*). The second and the third stages (described in Sec. 3) are each other independent and can be performed in parallel, which removes the event discovery runtime overhead. These three stages are iterated, so to adapt $\Phi(\cdot)$ (and $Y$) to the observation distribution change. The full algorithm is described in Alg. 1.

### 4.1 IMPLEMENTATION DETAILS

The weights $\phi$ and $\theta$ are initialized with a (semi-) orthogonal matrix, as described in (Saxe et al., 2014), apart from the head of $\Phi$ (i.e., the fully-connected layer on top of the CNN, see Fig. 1), which is initialized randomly. All the biases are initialized with zeros.

Following (Ji et al., 2019), to improve robustness, the head of $\Phi$ is duplicated 5 times with different initial random parameters. Moreover, $\Phi$ includes an auxiliary overclustering head with $C \times 5$ clusters. Each head produces a different representation with a potentially different degree of utility for the agent. A possible solution is the simple concatenation of the 5 different event probability

distributions $\Phi_1(o), ...\Phi_5(o)$. However, this solution would increase the dimensionality of $H(t)$. We instead select the "best" head using the MI value as the selection criterion as follows. We split training in two phases. In the first phase ($N * p_{pretrain}$ iterations of the outmost loop in Alg. 1) the agent receives only $I(t)$ as input and $\theta$ and $\phi$ (and each head in $\Phi$) are updated as usual (lines 8 and 9 in Alg. 1). Then, the head corresponding to the highest MI value (averaged over the last iterations) is selected and kept fixed during the second phase, in which the agent receives the full input ($H(t)$ included).

In all our experiments we use the same set of hyperparameters and the same network architecture. We adopt the training procedure and the hyperparameters in (Schulman et al., 2017), including the number of time steps equal to 10M. Our agent network architecture is similar to (Schulman et al., 2017), except from the additional history input ($H(t)$). $\Phi$ is composed of a CNN and a fully-connected layer (Sec. 3). Specifically, its CNN configuration is similar to $\Psi$, except the first layer, which takes only one channel as input. In Appendix A we list all other hyperparameters.

## 5 EXPERIMENTAL EVALUATION

We evaluate EDHR on two challenging deep RL benchmarks: the Arcade Learning Environment (Bellemare et al., 2012) (ALE) and the Obstacle Tower (Juliani et al., 2019). Moreover, we compare the results of EDHR with PPO and PPO-RNN, where the latter is described below.

**PPO-RNN.** $\Psi$ is the same as in EDHR and in PPO (Schulman et al., 2017). However, past observations are represented as follows. The encoder $\Phi$ is replaced with a similar network ($\Upsilon$) but without the softmax layer and having 256 neurons as output of the fully-connected layer instead of $C$. For each $t' \in \{t, ..., t - S + 1\}$, $\omega_t = \Upsilon(o_{t'})$ is input to a GRU (Cho et al., 2014) with a hidden layer of size 512 (re-initiated with zeros each $S$ steps). Finally, the hidden layer activation vector represents the history and replaces $H(t)$ in Fig. 2, keeping all the rest of the agent unchanged. Note that no self-supervision is used here to train the network components.

### 5.1 ENVIRONMENTS

In ALE we use 7 classic Atari 2600 video games: MsPacman, SpaceInvaders, Breakout, Gravitar, Qbert, Seaquest and Enduro. Empirically, balanced complexity and diverse game dynamics make this subset of the benchmark representative for comparison. In all these environments, $o_t$ is an RGB image. The action space $A$ is discrete and the number of actions varies in each game from 4 to 18. Following closely the experimental protocol in (Schulman et al., 2017), we apply the same observation pre-processing (frame downscaling, etc.).

Obstacle Tower is a 3D 3-rd person game environment. It consists of a number of floors, each floor being composed of several rooms. The layout is procedurally generated based on an environment seed. The rooms can contain puzzles to solve, the complexity of which increases with the floor number, creating a natural curriculum learning for the agent. The environment has high visual fidelity, realistic lighting, a variety of textures and geometry, physics-driven interactions. Possible actions include movements in four directions, camera rotations and jumps. The agent receives a reward of +1.0 after passing each floor and an auxiliary reward of +0.1 after solving an intermediate puzzle. An observation $o_t$ is an $84 \times 84$ RGB image, converted to grayscale. The action space is discrete, with 54 actions, which are reduced to 12 after removing redundancy. This environment is especially challenging because of several reasons. Unlike the Atari games with simple sprite-based graphics, this environment has diverse and realistic observations. The sparse reward signal complicates training and favours a low-dimensional agent's input. The agent is trained using environment seeds 0-7 and it is evaluated with seeds 50-57, which shows the generalisation abilities of the algorithm (see the *Train* and *Test* rows in Tab. 2, respectively).

### 5.2 RESULTS

In all the experiments we report the final scores averaged over 100 episodes. Each experiment is performed 3 times with a different random seed. We report the average value and the standard deviation. Tab. 1-2 show the results. "PPO (Sch.)" refers to the results reported in (Schulman et al., 2017), while the results "PPO" are based on our reproduction of that algorithm. We compare the

Table 1: Scores on the ALE benchmark

|  | PPO (Sch.) | PPO | PPO-RNN | EDHR (Ours) |
|---|---|---|---|---|
| MsPacman | 2096.5 | $1727.00 \pm 383.81$ | $2040.63 \pm 104.88$ | $\mathbf{2431.33} \pm 216.26$ |
| SpaceInvaders | 942.5 | $912.63 \pm 5.20$ | $952.30 \pm 126.78$ | $\mathbf{1089.48} \pm 155.26$ |
| Breakout | 274.8 | $\mathbf{413.07} \pm 22.12$ | $410.40 \pm 2.21$ | $368.24 \pm 36.54$ |
| Gravitar | 737.2 | $581.17 \pm 23.01$ | $\mathbf{600.33} \pm 197.36$ | $587.50 \pm 46.29$ |
| Qbert | 14293.3 | $11340.75 \pm 1463.59$ | $9388.58 \pm 2408.55$ | $\mathbf{11816.67} \pm 1952.70$ |
| Seaquest | 1204.5 | $1227.73 \pm 481.22$ | $1028.67 \pm 125.69$ | $\mathbf{1342.13} \pm 339.85$ |
| Enduro | 758.3 | $539.75 \pm 361.81$ | $499.00 \pm 432.16$ | $\mathbf{746.69} \pm 45.93$ |

Table 2: Scores on the Obstacle Tower benchmark

|  | PPO | PPO-RNN | EDHR (Ours) |
|---|---|---|---|
| Train | $1.54 \pm 0.07$ | $\mathbf{3.34} \pm 0.45$ | $2.24 \pm 0.18$ |
| Test | $1.35 \pm 0.24$ | $\mathbf{2.46} \pm 0.11$ | $1.52 \pm 0.23$ |

results with our reproduction to remove the impact of minor technical implementation details, but in Tab. 1 we also report the original results "PPO (Sch.)" as a reference. "PPO-RNN" is described above.

These results show the benefit of our history representation which is able to significantly boost the scores in most of the games, although more reactive games, like Breakout, do not show any benefit. PPO-RNN underperforms our EDHR on ALE benchmark. Note that, in most of the games, PPO-RNN has a similar or even *worse* performance of plain PPO. We believe this is due to the difficulty of training an RNN simultaneously with the agent in an RL framework, especially when the training budget is limited. Specifically, the training budget, adopted in all our experiments and uniformly with all the tested methods, consists of 10M of time steps (Sec. 4.1), which is equivalent to the number of observations $o_t$ an agent receives from the environment, considering all the parallel actors and all the iterations. Moreover, if we consider the wall-clock time (and with the same training budget), in all our experiments PPO-RNN is on average about two times *slower* then our EDHR. In Appendix B we include additional experiments with different hyperparameters of EDHR, an ablation model without self-supervision and another common RNN-based model.

The high score of PPO-RNN in Tab. 2 shows the importance of past information in the Obstacle Tower environment, where instantaneous observations are not sufficiently informative. EDHR successfully exploits the discovered events outperforming PPO. However, in this game our method underperforms PPO-RNN, indicating that in some cases the compact history representation can loose useful information.

The higher efficiency of our EDHR with respect to the training budget is shown in Fig. 3, where, for each game, we plot the scores reached by each method over the training iterations. These plots show that EDHR effectively makes use of the knowledge acquired in the unsupervised training stages and that, using an event-based history representation, most of the times training is easier. We believe this is due to the fact that the agent, in making a decision, can use the events, recorded in $H(t)$, as a sort of "summary" of the important information in a past trajectory.

## 5.3 DISCOVERED EVENT ANALYSIS

**What type of events does the method discover?** To answer this question, we visually examined the observations and the corresponding events of a trained agent in two environments, MsPacman and Obstacle Tower. Fig. 4 shows how the events are distributed during an episode of MsPacman. Specifically, for each time step $t$, we plot the predominant event $c_t^*$ corresponding to:

$$c_t^* = \arg\max_{c \in Y} \Phi_c(o_t) = \arg\max_{c \in Y} P(z = c | o_t) \tag{1}$$

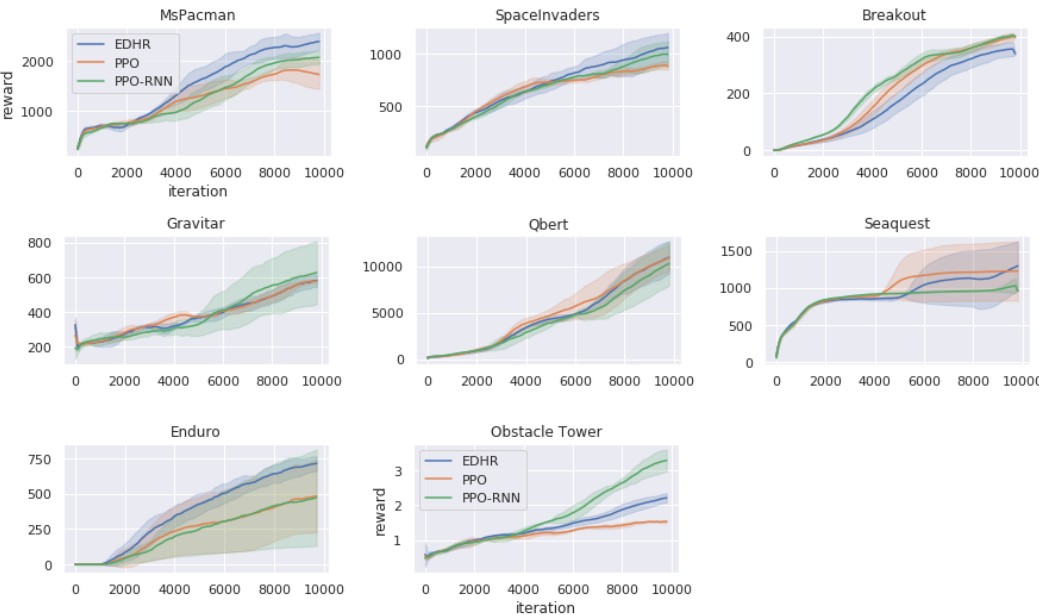

Figure 3: Training dynamics, smoothed for readability

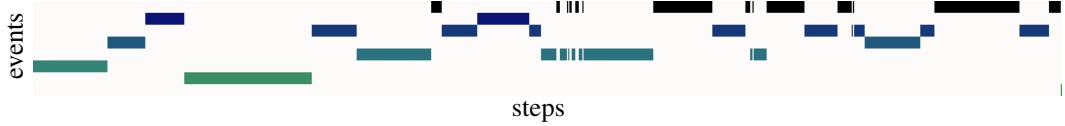

Figure 4: The event dynamics during an episode of MsPacman. Each color corresponds to a different predominant event computed using Eq. 1. For simplicity, the y-axis of the figure contains only a subset of the $C = 16$ events.

This figure shows that the predominant event tend to be constant for a while during an episode, which is coherent with the expectation that important information about the past (an "event") changes with a low frequency. Moreover, we observed that these events correspond to specific changes of the environment. For instance, when the agent collects an object called "Power Pellet", which produces a switching of the color of the ghosts to blue and it allows these ghosts to be eaten by the agent, then at that moment there is a predominant event change in $H(t)$.

Figure 5 shows some observations and the corresponding predominant events extracted from an episode of the Obstacle Tower environment. The observations have a sequential order (but they are not each other time-adjacent). New events correspond to the transition to a new room, or to different viewpoints of the same room. As shown in this figure, in the middle of the trajectory, the agent observes a green door, and when it later returns to it, the event changes back to the corresponding value. In Appendix D we plot another episode of the same environment, with high-frequency selected observations and the corresponding predominant events, which more clearly shows the semantics behind the clusters discovered by the encoder and used by the agent.

**Sparsity of our history representation**. Using Obstacle Tower we computed the probability that, in a trained agent, a random element in the $S \times C$ matrix $H(t)$ has a value $q$ such that $0.1 < q < 0.9$. This probability is 0.00146. Using the range of values $0.01 < q < 0.99$, the probability is 0.0031. These empirical results show that our representation is very sparse and "binary", meaning that all the information is concentrated around the values 0 and 1.

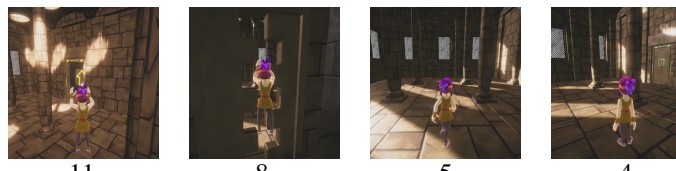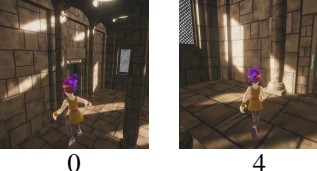

| 11 | 8 | 5 | 4 | 0 | 4 |

Figure 5: Obstacle Tower predominant events associated with sequentially ordered observations of an episode.

## 6 CONCLUSION

We presented a method for history representation in RL which is an alternative to common solutions based on RNNs. Our EDHR is based on the idea that important information about the past can be "compressed" using events. Specifically, these events are automatically discovered using a modification of the ICC clustering method (Ji et al., 2019), which is performed jointly with the agent training and iterated through time, to adapt the discovered events to the new observations. In EDHR, visual information is represented using two different networks: $\Phi$ and $\Psi$. The latter is trained using a reward signal, so it presumably extracts task-specific information from the observations. On the other hand, the encoder $\Phi$ is trained using self-supervision, and thus it focuses on patterns which are repeated in the data stream, potentially leveraging a larger quantity of supervision signal. Although self-supervision has been explored in other RL and non-RL works, this is the first work to show how the discovered information can be exploited in the form of discrete events used for history representation.

Our results, based on the challenging deep RL benchmarks, ALE and Obstacle Tower, show that EDHR can more effectively represent information about the past in comparison with task-oriented representations such as those used in PPO and PPO-RNN.

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

Table 3: Hyperparameters

| | |
|---|---|
| Number of clusters $C$ | 16 |
| History size $S$ | 32 |
| Temporal window $L$ | 3 |
| Number of epochs $n_E$ | 10 |
| Batch size (to update $\phi$) | 512 |
| Learning rate (to update $\phi$) | .0005 |
| Pretraing iterations ratio $p_{pretrain}$ | 15% |

Table 4: Additional experiments

| | MsPacman | SpaceInvaders | Breakout |
|---|---|---|---|
| EDHR $L = 3, S = 32$ (default) | $2431.33 \pm 216.26$ | $\mathbf{1089.48} \pm 155.26$ | $368.24 \pm 36.54$ |
| EDHR $L = 1, S = 32$ | $\mathbf{2473.87} \pm 355.57$ | $1048.72 \pm 128.57$ | $375.54 \pm 31.90$ |
| EDHR $L = 8, S = 32$ | $2038.33 \pm 435.01$ | $910.85 \pm 44.65$ | $389.07 \pm 6.31$ |
| EDHR $L = 3, S = 16$ | $2148.03 \pm 247.37$ | $864.05 \pm 122.53$ | $411.09 \pm 11.89$ |
| EDHR $L = 3, S = 64$ | $1916.67 \pm 76.67$ | $875.52 \pm 58.46$ | $373.30 \pm 9.49$ |
| Without self-supervision | $1897.70 \pm 90.54$ | $981.20 \pm 171.63$ | $\mathbf{411.58} \pm 2.86$ |
| PPO-RNN without $I(t)$ | $1317.47 \pm 136.67$ | $746.80 \pm 33.82$ | $189.40 \pm 29.84$ |

## A  HYPERPARAMETERS

In Tab. 3 we list all our hyperparameters which are different from (Schulman et al., 2017).

## B  ADDITIONAL EXPERIMENTS

In Tab. 4 we use three environments of the ALE benchmark to show the impact of different EDHR hyperparameter choices together with an ablation study. Using a minimum length temporal window ($L = 1$) has a similar performance to $L = 3$, whilst a much larger value $L = 8$ degrades the performance. This is probably due to the fact that a large $L$ value blurs out the borders between clusters, thus the event assignment is more ambiguous, less stable and less useful for the agent training.

As expected, a short history ($S = 16$) degrades the performance for the MsPacman and the SpaceInvaders environments. However, in this case Breakout performs slightly better, demonstrating that in a reactive environment the agent focuses on the instantaneous representation and the shorter history stimulates this. On the other hand, a longer history ($S = 64$) degrades the score for all the environments: distant events are less useful for the training, whereas the doubled size of $H(t)$ makes the optimisation harder.

To show the impact of self-supervision, we include in Tab. 4 a variant of EDHR without the event discovery stage. Specifically, "Without self-supervision" is equal to EDHR, but $\Phi$ is trained using the gradient of the reward signal, jointly with $\Psi$. This experiment shows a dramatic degradation in the MsPacman and the SpaceInvaders environments, which highlights the gain which can be obtained using the proposed additional self-supervision. However, as above observed, in the more reactive environment Breakout the past information is less important, and the score achieved by "Without self-supervision" is equal to the plain PPO (Tab. 1), which indicates that the model ignores the history.

Finally, the last row in Tab. 4 concerns a variant of the PPO-RNN algorithm, where the input $I(t)$ and the corresponding CNN $\Psi$ are removed. In this case the network receives the last $S$ observations of the environment as images, each image is processed using the CNN, the obtained features are input to the RNN, and the network is trained using the PPO algorithm. This model is similar to the common RNN-based solutions (e.g. Mnih et al. (2016)). It is trained using the same hyperparameters of the

other experiments. Empirically we observed that this solution is prone to a higher variance in the CNN parameter updates, requiring larger training budget.

## C  INVARIANT INFORMATION CLUSTERING

In this section we briefly describe the self-supervised method we used for clustering, referring the reader to the original paper (Ji et al., 2019) for more details. Let $x, x' \in X$ be two different images sharing similar semantic information. In our case, $x, x'$ are two temporally-close frames, both contained in the temporal translation window (Sec. 3), while in (Ji et al., 2019) $x, x'$ represent the same image with some random perturbations (e.g. scaling, rotation, color adjustment, etc.).

$\Phi$ is a network mapping an image into a probability distribution $\Phi : X \rightarrow [0, 1]^C$, where $C$ is the number of clusters (in our case, events). $\Phi(x)$ is the probability distribution of a discrete random variable $z$ which (soft-)assigns $x$ to the elements in $Y = \{1, ..., C\}$ and can be defined as: $\Phi_c(x) = P(z = c|x)$. If $x$ and $x'$ share the same semantic content, then they should be classified equally, otherwise they should be assigned to different clusters. This objective is obtained by maximizing the Mutual Information between the corresponding representations:

$$\max_{\Phi} I(\Phi(x), \Phi(x')).  \qquad (2)$$

In more detail, a pair $(x, x')$ corresponds to the two variables $(z, z')$, with conditional joint distribution $P(z = c, z' = c'|x, x') = \Phi_c(x) \cdot \Phi_{c'}(x')$, where the equality holds when $z$ and $z'$ are independent (Ji et al., 2019). After marginalization over the batch, the joint probability distribution is given by the $C \times C$ matrix $\mathbf{P}$, where each element at position $c, c'$ is $\mathbf{P}_{cc'} = P(z = c, z' = c')$:

$$\mathbf{P} = \frac{1}{n} \sum_{i=1}^{n} \Phi(x_i) \cdot \Phi(x_i')^{\intercal}.  \qquad (3)$$

The marginals $\mathbf{P}_c = P(z = c)$ and $\mathbf{P}_{c'} = P(z' = c')$ correspond to the sums over the rows and the columns of $\mathbf{P}$, rispectively. Finally, the optimisation objective in Eq. 2 is computed directly using the definition of Mutual Information:

$$I(z, z') = I(\mathbf{P}) = \sum_{c=1}^{C} \sum_{c'=1}^{C} \mathbf{P}_{cc'} \cdot ln \frac{\mathbf{P}_{cc'}}{\mathbf{P}_c \cdot \mathbf{P}_{c'}}.  \qquad (4)$$

## D  EVENT VISUALIZATION

Figure 6 shows a detailed sequence of observations and the corresponding predominant events from an episode of Obstacle Tower. We observe that some events correspond to semantically general classes and do not focus only on visual appearance. For instance, event 15 indicates when the agent passes from one room to another, despite each transition (room) is visually different. Events 2 and 12 correspond to specific rooms, event 8 detects the door, events 4 and 14 indicates specific views of a room.

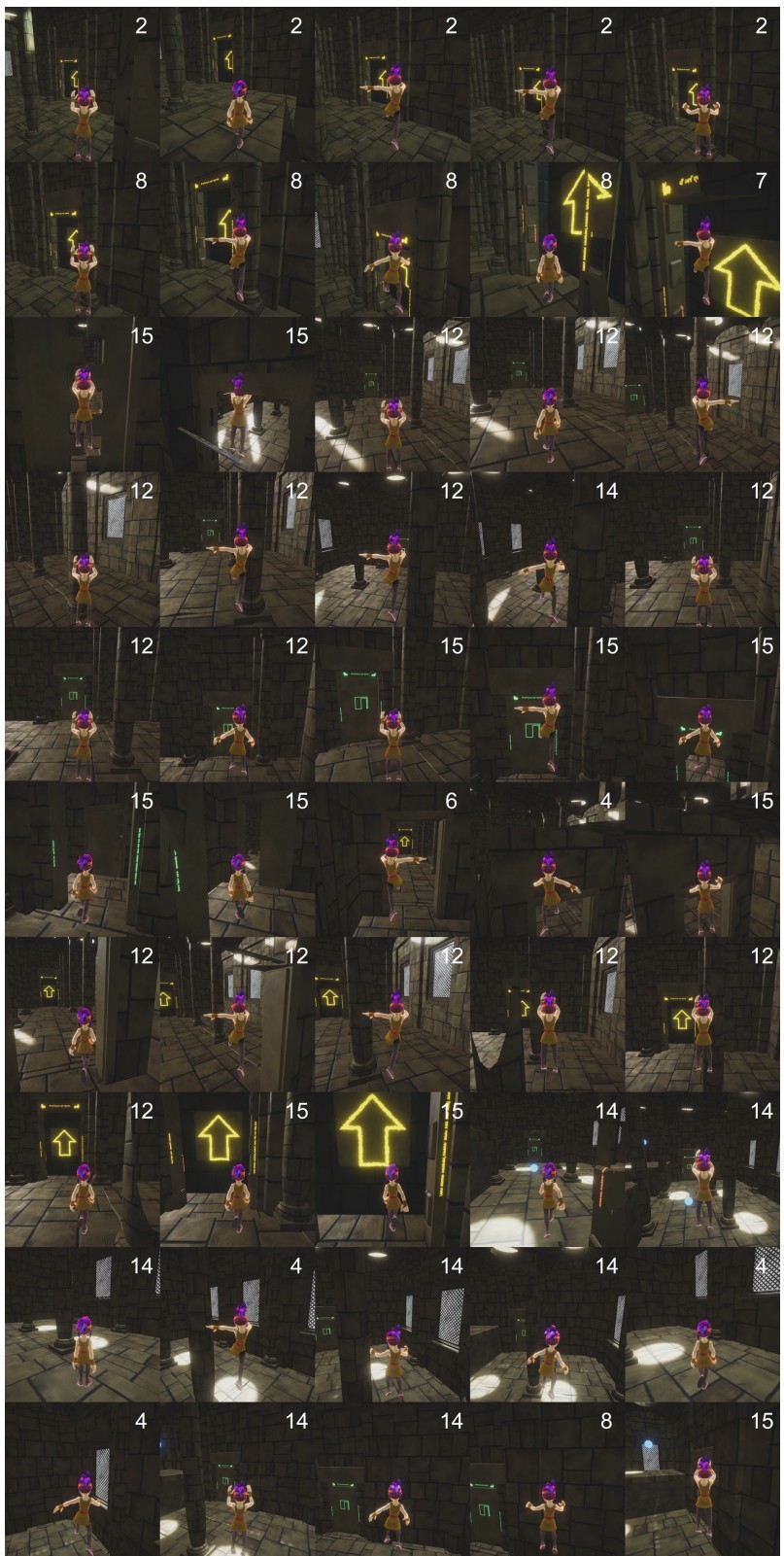

Figure 6: Detailed sequence of Obstacle Tower events

