# OpenReview forum: "Event Discovery for History Representation in Reinforcement Learning"
_ICLR.cc/2020/Conference — Reject_

### Official Review · AnonReviewer1 · 2019-10-05
**Official Blind Review #1**

**Rating:** 6

**Review:**

The authors study the problem of RL under partially observed settings. While most current (D)RL approaches use RNNs to tackle this problem, RNNS are trickier to optimise than FFNNs - in practice RNN-based DRL agents can perform well on partially observed problems, may require more effort to optimise, and may underperform FFNNs on domains with no/less partial observability. The proposed solution is to use a FFNN, but provide a "history representation", which is a set of feature vectors for previous timesteps that is extracted from a second network. The second network is trained separately using self-supervision (specifically, IIC, but adapted to use temporal consistency of observations rather than data augmentation). The proposed algorithm outperforms both PPO with a FFNN and PPO with an RNN on 5/7 Atari games (mainly games where partial observability is higher), as well as on the new and challenging Obstacle Tower benchmark - though PPO with RNN results are conspicuously missing on the latter! Given the promising approach (which also has a 2x better wall-clock training time than PPO with an RNN) and results, I would give this paper a weak accept. Some nice properties are that the instantaneous feature extractor is trained using RL, while the history feature extractor is trained using self-supervision at a high level (not pixel level), so that they are probably complementary; in addition it appears that in practice the resulting history features are sparse and usually binary, which is an intriguing and potentially useful property for future work.

There are several things to be done to improve the paper however. First and foremost, the results should be run over several seeds with standard deviation/error reported (it appears that this might be the case for Obstacle Tower, but no error is reported). I believe that the large improvements in some of the domains are significant, but it would be best to have this confirmed empirically. The authors could improve the presentation of background material by giving techniques names instead of using just author names. Although it would be expensive to show how changing L affects performance on all domains, some quantitative results on this hyperparameter would be useful - the same applies to C. The authors should also provide more clarity on choosing the history head - is the head identity fixed after pretraining? It would be useful to know how the more standard PPO with RNN architecture performs, but the authors' choice of architecture is a fitting comparison to their method, and does perform well in practice on the more partially observed domains, so the current setup is satisfactory.

**Experience Assessment:**

I have published one or two papers in this area.

**Review Assessment: Checking Correctness Of Derivations And Theory:**

N/A

**Review Assessment: Checking Correctness Of Experiments:**

I carefully checked the experiments.

**Review Assessment: Thoroughness In Paper Reading:**

I read the paper thoroughly.

---

> ### Author Response · Authors · 2019-11-15
> **question 1**
>
> "First and foremost, the results should be run over several seeds with standard deviation/error reported (it appears that this might be the case for Obstacle Tower, but no error is reported). I believe that the large improvements in some of the domains are significant, but it would be best to have this confirmed empirically."
>
> We performed all experiments with 3 random seeds and we have updated the results in Sec. 5.2 (Tab. 1, 2) in the new version reporting the mean and the standard deviation value.

---

> ### Author Response · Authors · 2019-11-15
> **question 2**
>
> "Although it would be expensive to show how changing L affects performance on all domains, some quantitative results on this hyperparameter would be useful - the same applies to C."
> ----
> We include additional experiments in the appendix of the new version for the hyperparameters L and S (see also answer 5 to Reviewer 2).

---

> ### Author Response · Authors · 2019-11-15
> **question 3**
>
> "The authors should also provide more clarity on choosing the history head - is the head identity fixed after pretraining?"
>
> Yes, the head identity is kept fixed after pretraining, we wrote it more explicitly in the updated version (Sec. 4.1).

---

> ### Author Response · Authors · 2019-11-15
> **question 4**
>
> "It would be useful to know how the more standard PPO with RNN architecture performs, but the authors' choice of architecture is a fitting comparison to their method, and does perform well in practice on the more partially observed domains, so the current setup is satisfactory."
>
> Additionally we added two models with experiments in 3 Atari environments in the appendix of the new version.

---

### Official Review · AnonReviewer2 · 2019-10-23
**Official Blind Review #2**

**Rating:** 3

**Review:**

This paper proposes a new way to represent past history as input to an RL agent, that consists in clustering states and providing the (soft) cluster assignment of past states in the input. The clustering algorithm comes from previous work based on mutual information, where close (in time) observations are assumed to be semantically similar. The proposed scheme, named EDHR (Event Discovery History Representation), is shown to perform better than PPO and an RNN variant of PPO on (most of) 7 representative Atari games, and better than PPO on the Obstacle Tower benchmark.

I would like to see this paper eventually published as I find the proposed technique original and quite relevant to current RL research, however I feel like its empirical evaluation is too weak at this time, which is why I am recommending rejection. I hope the results can be strengthened in a revised version so that I can increase my rating.

The main limitations of the current empirical evaluation are:
•	Only 7 Atari games are used (vs 49 in the PPO paper the proposed technique is compared to), without justification for how they were chosen, and it seems like only 1 run is performed on each game (while RL algorithms are well known to exhibit high variance)
•	On Obstacle Tower there seems to be also only one run of each algorithm (more runs could be done with different training & testing seeds in order to get an idea of the variance)
•	There is no comparison to PPO+RNN on Obstacle Tower
•	I think a natural and important baseline to compare to is using the same architecture as in Fig. 2 but where the mapping Phi(o_t) is learned through regular backprop (using the same loss as when learning the mapping I(t)). This would validate that the advanced self-supervised clustering technique from Ji et al. (2018) is actually useful, and thus that the observed improvements are not simply due to providing 32 frames of history vs. 4 as in vanilla PPO.

Other (more minor) remarks:
•	Please explain better how the clustering technique from Ji et al. (2018) works, possibly in the Appendix if there is no room in the main body of the paper. This will make the paper more self-contained.
•	Without fully understanding how this clustering technique works, it is difficult to me to get an intuition on how the clusters evolve during training, especially as new types of states are discovered by the agent. Some discussion on this topic would be appreciated.
•	It would also be interesting to analyze the impact of varying the various new hyper-parameters (in particular S, C and L)
•	On the first line of the last paragraph on p. 4, there is a missing reference « (Sec. ) »
•	Overall there are a bunch of typos throughout the paper that could easily be fixed


Follow-up on author response: I remain inclined to stick to my rejection recommendation. I appreciate the efforts in providing more results, but I find them too limited and not entirely convincing: Table 4 is only done on 3 games, and the only one where the proposed method has a clear edge over "Without self-supervision" is MsPacMan. In addition, the fact that PPO+RNN shows much better performance than the proposed method on Obstacle Tower is also worrying.

**Experience Assessment:**

I have read many papers in this area.

**Review Assessment: Checking Correctness Of Derivations And Theory:**

N/A

**Review Assessment: Checking Correctness Of Experiments:**

I carefully checked the experiments.

**Review Assessment: Thoroughness In Paper Reading:**

I read the paper thoroughly.

---

> ### Author Response · Authors · 2019-11-15
> **question 1**
>
> "Only 7 Atari games are used (vs 49 in the PPO paper the proposed technique is compared to), without justification for how they were chosen, and it seems like only 1 run is performed on each game (while RL algorithms are well known to exhibit high variance)
> On Obstacle Tower there seems to be also only one run of each algorithm (more runs could be done with different training & testing seeds in order to get an idea of the variance)"
>
> In new version in Sec. 5.1 we added motivation for choosing this subset on the Atari benchmark. We performed all experiments with 3 random seeds and we have updated our experimental results in Sec. 5.2 (Tab. 1, 2) reporting the mean and the standard deviation values.

---

> ### Author Response · Authors · 2019-11-15
> **question 2**
>
> "There is no comparison to PPO+RNN on Obstacle Tower"
>
> We performed this experiment, whose results have been included it in the new version of the paper (see Sec. 5.2, Tab. 2 and also the answer number 11 to Reviewer 3).

---

> ### Author Response · Authors · 2019-11-15
> **question 3**
>
> "I think a natural and important baseline to compare to is using the same architecture as in Fig. 2 but where the mapping Phi(o_t) is learned through regular backprop (using the same loss as when learning the mapping I(t)). This would validate that the advanced self-supervised clustering technique from Ji et al. (2018) is actually useful, and thus that the observed improvements are not simply due to providing 32 frames of history vs. 4 as in vanilla PPO."
>
> Thank you for your suggestion: it's an interesting idea. We implemented this solution and we performed an additional experiment, reported in the appendix of the new version.

---

> ### Author Response · Authors · 2019-11-15
> **question 4**
>
> "Please explain better how the clustering technique from Ji et al. (2018) works, possibly in the Appendix if there is no room in the main body of the paper. This will make the paper more self-contained.
> Without fully understanding how this clustering technique works, it is difficult to me to get an intuition on how the clusters evolve during training, especially as new types of states are discovered by the agent. Some discussion on this topic would be appreciated."
>
> We added a brief description of IIC in the appendix in the new version.
> The encoder is trained online with new observations, evolving with the agent during the training, as listed in Alg. 1.

---

> ### Author Response · Authors · 2019-11-15
> **question 5**
>
> "It would also be interesting to analyze the impact of varying the various new hyper-parameters (in particular S, C and L)"
>
> We included additional experiments in the appendix in the new version concerning parameters L and S. Unfortunately, due to limited rebuttal time, the experiments with different values of C are not ready yet. We will publish the results of the latter as soon as they are ready.

---

> ### Author Response · Authors · 2019-11-15
> **question 6**
>
> "On the first line of the last paragraph on p. 4, there is a missing reference « (Sec. ) »"
> "Overall there are a bunch of typos throughout the paper that could easily be fixed"
>
> We fixed the typos in the new version.

---

### Official Review · AnonReviewer3 · 2019-10-28
**Official Blind Review #3**

**Rating:** 1

**Review:**

The paper focuses on the problem learning state representations that can effectively capture historical information
in POMDPs. Specifically, the paper proposes an alternative approach to using RNN's for capturing such history - the authors adopt a variant of recently proposed Invariant Information Clustering approach to discover important events in past observation and then use these events to learn a probability distribution over observations to represent the state information. The goal here is to address the unstable and inefficient training issues associated with RNN based architectures. The authors validate their approach with experiments on seven tasks on Atari 57 benchmark and Obstacle Tower, both with discrete action space and compare their performance against both original and RNN versions of PPO.

The paper addresses an interesting problem to help learning better state representation in absence of  complete information about the environment. The approach of using IIC clustering (or any clustering approach) for learning state representations is novel. The overall goal of replacing RNN based methods in order to achieve stable, efficient learning in presence of budget constraints is very useful and hence this approach is potentially a good step in that direction. Although not adequate, the results in figure 3 provides good insight into effectiveness of the method in making training easier.

However, I am inclining to reject this paper for the following reasons:
(1) The motivation for using proposed clustering approach for history representation is neither clear  and nor well exposed. MI based methods are inherently difficult to learn and hence this approach needs  rigorous analysis on why it works when it does and how it fails.
(2) The paper fails to position the new approach in comparison to related works both in discussions and experiments.
(3) The experiments are very limited in nature and fails to demonstrate the efficacy of the proposed approach effectively. Further, the key contribution focusing on learning effective representations under constrained budget is not adequately tested.

Major concerns:

Motivation
-------------
- It is not clear how the proposed clustering mechanism to discover events allows to successfully capture information that an RNN based approach does. For instance, RNN helps to capture long-term dependencies but it is very hard to interpret the proposed model from that aspect. Also, why is this particular clustering approach (Invariant Information Clustering) chosen? The motivation for using this approach is not clear and overall combination appears adhoc.
- Further, RNN based methods can retain order information in sequential history of observations. However,  this method appears to not consider that information. Is this is true or am I mistaken here? If this is true, why would this not create issues in learning good representations for task where order is indeed important?
- The paper discusses several articles that provide background on the methods used however it fails to position the exposition in comparison to existing RNN based approaches which is a big miss as the goal of the paper is to replace RNN based approaches [1,2,3,4].

Method
------
- In the event discovery stage, it is not clear why using consecutive observations is not useful. Also, if one does use L=1 (consecutive observations), can one reduce this method to RNN as you end of capturing all the previous history? Also, why L=3 is good across all different tasks? does it have any relation with the use of 4 frames in I(t)?
- The authors mention tat H(t) matrix is highly sparse but also low-dimensional in all their experiments.  Would this is be the case for any other task? If not, is this a limitation of the method that you need S and C  to be small?
- The authors attempt to use clustering based approach with the hope of recovering important information, however
mention that H(t) stores all the past events. Isn't this contradictory? Also, why can RNN with an attention based mechanism not achieve similar effect?
- It is also useful to analyse how will this method work in presence of long vs short history? Will the clustering itself and hence the learned representations get affected by length of available history?

Experiments
----------
- Focusing only on PPO and designing an RNN version of PPO constrains the effectiveness of experiments in validating the approach. Authors mention difficulty of training with RNN as one reason for PPO with RNN's under performance but this is not convincing Could the performance of PPO with RNN be limited only due to specific RNN architecture used?
- Authors must compare with other methods that use RNN approaches (e.g. [1]). Also, if methods such [2],[3],[4] are not  directly applicable, they must atleast, use their RNN based architectures to modify PPO and compare several baselines.
- Why do the authors not report PPO with RNN for Obstacle Tower?
- For the experiments, authors use a specific 10000 steps budget but this seems to be highly curated. The experiments
would be stronger if the authors show experiment over a range of budget. This will also give insights on when does RNN becomes better and is there a budget after which both methods perform equally well or RNN based approaches
surpass the current approaches.
- Figure 3: Training dynamics seem to favor RNN approach for BreakOut, Gravitar. Do the authors have insight on why
this is the case?

Minor points to improve submission not affecting the score:

- The paper needs to be proofread for various typos and sentence construction issues
- Table 1: For Qbert, original PPO (Sch.) is best performing, not EDHR
- [5] talks about difficulties in MI based methods and I encourage the authors to look at the analysis in
this paper. As it is only an arXiv version and not published yet, I have not based my assessment on the
existence of this paper but still connection to such analysis will make this paper stronger.


[1] Deep Variational Reinforcement Learning for POMDPs, Igl et. al.
[2] On improving deep reinforcement learning for POMDPs, Zhu et. al.
[3] Policy Learning with continuous memory states in partially observed robotic control, Zhang et. al.
[4] Memory-based control with recurrent neural networks, Heess et. al.
[5] On Mutual Information Maximization and Representation Learning, Tschannen et. al.

**Experience Assessment:**

I have read many papers in this area.

**Review Assessment: Checking Correctness Of Derivations And Theory:**

N/A

**Review Assessment: Checking Correctness Of Experiments:**

I carefully checked the experiments.

**Review Assessment: Thoroughness In Paper Reading:**

I read the paper thoroughly.

---

> ### Author Response · Authors · 2019-11-15
> **question 1**
>
> "It is not clear how the proposed clustering mechanism to discover events allows to successfully capture information that an RNN based approach does. For instance, RNN helps to capture long-term dependencies but it is very hard to interpret the proposed model from that aspect. "
>
> In our proposed history representation (H), each observation is represented as a distribution over a set of discovered events. Basically, H is a record of the past S events. And events describe “landmarks” of the environment (see Sec. 1). Hence, H represents past information with respect to this dictionary of landmarks. In contrast, RNN represents past information which can be prone to forgetting and it is not necessarily related to important environment landmarks/events.

---

> ### Author Response · Authors · 2019-11-15
> **question 2**
>
> "Also, why is this particular clustering approach (Invariant Information Clustering) chosen? The motivation for using this approach is not clear and overall combination appears adhoc."
>
> We use temporally-close frames as positive pairs for IIC, thus the obtained clusters are consistent in time (see Sec. 5.3), which is required by the definition of “event” (Sec. 1). This property, jointly with the high efficiency of IIC, make the IIC method a proper candidate for our approach.

---

> ### Author Response · Authors · 2019-11-15
> **question 3**
>
> "Further, RNN based methods can retain order information in sequential history of observations. However,  this method appears to not consider that information. Is this is true or am I mistaken here? If this is true, why would this not create issues in learning good representations for task where order is indeed important?"
>
> The event order is explicitly retained in our method: each event has a specific row in the history matrix H (see Fig. 2 and Sec. 3).

---

> ### Author Response · Authors · 2019-11-15
> **question 4**
>
> "The paper discusses several articles that provide background on the methods used however it fails to position the exposition in comparison to existing RNN based approaches which is a big miss as the goal of the paper is to replace RNN based approaches [1,2,3,4]."
>
> Method [1] is interesting, we included it in the new version in the "Related Work", describing the difference with our method. Additionally, in the Appendix we included experiments with a common RNN-based architecture. Method [2] is similar to (Hausknecht &  Stone (2015)) presented in "Related Work". Methods [3,4] focus on memory for RL, which is a slightly different direction of research, and these methods were not demonstrated on complex environments like the Atari benchmark, making the comparison with our approach infeasible.

---

> ### Author Response · Authors · 2019-11-15
> **question 5**
>
> "- In the event discovery stage, it is not clear why using consecutive observations is not useful. Also, if one does use L=1 (consecutive observations), can one reduce this method to RNN as you end of capturing all the previous history? Also, why L=3 is good across all different tasks? does it have any relation with the use of 4 frames in I(t)?"
>
> Consecutive observations are useful but may be less informative than farther observations, being the latter more different to each other. In the appendix of the new version we show experiments with L=1 and L=8. The L=1 case cannot be reduced to an RNN approach, because they are conceptually completely different methods: In a common RNN there is no self-supervision, and the goal of a recurrent network is (generally speaking) not to predict the future but to store the past. RNNs are used in some specific self-supervised methods (see Sec. 2), but when an RNN is used as input to an RL algorithm, than it usually refers to a network trained using the reward as the supervisory signal and without any future prediction step (Sec. 2). In contrast, in our method L defines a temporal translation window which is used in the self-supervised training stage and not in the representation of the past information input to the agent.
>
> For the same reason, the value of L is not related with the number of frames in the instantaneous representation: L is a hyperparameter of the unsupervised encoder \Phi, while the size of I(t) is a hyperparameter of the RL model.

---

> ### Author Response · Authors · 2019-11-15
> **question 6**
>
> "The authors mention tat H(t) matrix is highly sparse but also low-dimensional in all their experiments.  Would this is be the case for any other task? If not, is this a limitation of the method that you need S and C  to be small?"
>
> The sparsity (which is obtained in any environment) is a property of the IIC method (specifically because of the last softmax layer). In more detail, the probability distribution of an observation with respect to a set of events is very peaked, meaning that each step is represented with C-1 numbers close to zero and one number close to one. Thus the history matrix is represented with S x (C-1) numbers close to zero and S numbers close to one, meaning that the matrix is sparse. The dimensionality of the matrix is SxC.
> Note that this is not a limitation of the method, instead it is an advantage, because an even more compact history representation may be produced, if necessary (e.g., using an extra fully connected layer for a dimensionality reduction). Representing past step with a low-dimensional vector is an advantage because it can be processed by the RL agent using a smaller-capacity network.

---

> ### Author Response · Authors · 2019-11-15
> **question 7**
>
> "The authors attempt to use clustering based approach with the hope of recovering important information, however mention that H(t) stores all the past events. Isn't this contradictory? Also, why can RNN with an attention based mechanism not achieve similar effect?"
>
> Note that H(t) explicitly stores only the last S events, not all the past events (to avoid confusion, in Sec. 3 we changed the sentence “in H(t) all the past events are stored” with “in H(t) all the past S events are stored”, but we believe this should also be clear from the context). In other words, H(t) explicitly stores all the past S events observed by the agent, while an RNN automatically chooses what to store in its hidden state and what not.
>
> Note that the clustering process (“Event discovery stage”, Sec. 3) is separated from the “History Representation” stage. In the former, clustering is used to discover a dictionary of relevant events. In the latter, H represents the past S steps of the agent with respect to this dictionary. The dictionary should be representative of the whole environment, while H only represents the recent past.
>
> Concerning your proposed RNN +Attention based solution, you probably mean that Attention should be used to focus on specific (recent) inputs. However, as aforementioned, it is important to distinguish recent information (i.e., the past S agent observations) from the whole information used in the clustering process using self-supervision. With and Attention-based RNN, Attention can be used to select/emphasize some of the last S observations, but this largely differs from our “Event discovery stage”, where patterns of observations are extracted from all the past trajectories.

---

> ### Author Response · Authors · 2019-11-15
> **question 8**
>
> "It is also useful to analyse how will this method work in presence of long vs short history? Will the clustering itself and hence the learned representations get affected by length of available history?"
>
> Please, see our previous answers: the clustering process and the history construction are two separate phases (see Sec. 3, the “Event discovery stage” and the “History Representation” stage), so they are conceptually independent.
> We included additional experiments in the appendix of the new version to show the impact of the history length (S). Note that S is not a parameter of the clustering encoder \Phi, thus it does not have any effect on the representation learning.

---

> ### Author Response · Authors · 2019-11-15
> **question 9**
>
> "Focusing only on PPO and designing an RNN version of PPO constrains the effectiveness of experiments in validating the approach. Authors mention difficulty of training with RNN as one reason for PPO with RNN's under performance but this is not convincing Could the performance of PPO with RNN be limited only due to specific RNN architecture used?"
>
> As mentioned in the paper, our proposed history representation is independent of the specific RL approach. Good performance and technical simplicity make PPO a good candidate for demonstration. We design the PPO-RNN method to be as similar as possible to our EDHR for a fair comparison. We additionally include two other methods in the appendix of the new version.

---

> ### Author Response · Authors · 2019-11-15
> **question 10**
>
> "Authors must compare with other methods that use RNN approaches (e.g. [1]). Also, if methods such [2],[3],[4] are not  directly applicable, they must atleast, use their RNN based architectures to modify PPO and compare several baselines."
>
> Please, see our answer number 4 which concern the methods [1,2,3,4]. In addition, we included experiments using two other methods in the appendix of the new version.

---

> ### Author Response · Authors · 2019-11-15
> **question 11**
>
> "Why do the authors not report PPO with RNN for Obstacle Tower?"
>
> This experiment requires larger computational budget with respect to others, and this is the reason for which it was missing in the initial submission. Following your suggestion, we performed this experiment, whose results have been included it in the new version of the paper (Sec. 5.2, Tab. 2).

---

> ### Author Response · Authors · 2019-11-15
> **question 12**
>
> "For the experiments, authors use a specific 10000 steps budget but this seems to be highly curated. The experiments would be stronger if the authors show experiment over a range of budget. This will also give insights on when does RNN becomes better and is there a budget after which both methods perform equally well or RNN based approaches
> surpass the current approaches."
>
> We adopt the hyperparameter values (number of steps included) taken from the original PPO paper (Schulman et al., 2017) (see Sec. 4.1). Scalability of the method is an interesting direction for the research, we plan it for future work.

---

> ### Author Response · Authors · 2019-11-15
> **question 13**
>
> "Figure 3: Training dynamics seem to favor RNN approach for BreakOut, Gravitar. Do the authors have insight on why this is the case?"
>
> Performance in these environments is similar for PPO and PPO-RNN, indicating that history is less important, and PPO-RNN learns to focus on instantaneous observation. This is probably due to the fact that such environments are more reactive, so the salient information is contained in the very recent past and can be represented by the instantaneous information.

---

> ### Author Response · Authors · 2019-11-15
> **question 14**
>
> "Table 1: For Qbert, original PPO (Sch.) is best performing, not EDHR"
>
> Note that in that table we compare columns “PPO” (our reproduction of the PPO algorithm), “PPO-RNN” and “EDHR”. This is done to remove the impact of minor technical implementation details. Conversely, the column corresponding to “PPO (Sch.)“ was added only as a reference. In the new version of the paper we have better clarified this point in the beginning of Sec. 5.2 and we have graphically separated the “PPO (Sch.)” column from the other columns.

---

> ### Author Response · Authors · 2019-11-15
> **question 15**
>
> "[5] talks about difficulties in MI based methods and I encourage the authors to look at the analysis in this paper. As it is only an arXiv version and not published yet, I have not based my assessment on the existence of this paper but still connection to such analysis will make this paper stronger."
>
> Thank you for your suggestion, it's indeed an interesting paper and its analysis could improve the foundation of the IIC method. However, it's quite a different direction with respect to our work.

---

### Decision · Program_Chairs · 2019-12-19

**Decision:**

Reject

**Comment:**

The authors propose approaches to handle partial observability in reinforcement learning. The reviewers agree that the paper does not sufficiently justify the methods that are proposed and even the experimental performance shows that the proposed method is not always better than baselines.